# Production and characterization of antibody against *Opisthorchis viverrini* via phage display and molecular simulation

Sitthinon Siripanthong[1], Anchalee Techasen[2,3], Chanin Nantasenamat[4], Aijaz Ahmad Malik[4], Paiboon Sithithaworn[2,5], Chanvit Leelayuwat[6], Amonrat Jumnainsong[6]*

1 Graduate School, Khon Kaen University, Khon Kaen, Thailand, 2 Liver Fluke and Cholangiocarcinoma Research Institute, Khon Kaen University, Khon Kaen, Thailand, 3 Faculty of Associated Medical Sciences, Department of Clinical Microbiology, Khon Kaen University, Khon Kaen, Thailand, 4 Faculty of Medical Technology, Center of Data Mining and Biomedical Informatics, Mahidol University, Bangkok, Thailand, 5 Faculty of Medicine, Department of Parasitology, Khon Kaen University, Khon Kaen, Thailand, 6 Faculty of Associated Medical Sciences, The Centre for Research and Development of Medical Diagnostic Laboratories and Department of Clinical Immunology and Transfusion Sciences, Khon Kaen University, Khon Kaen, Thailand

* amonrat@kku.ac.th

**Data Availability Statement:** All relevant data are within the manuscript and its Supporting Information files.

## Abstract

In this study, a key issue to be addressed is the safe disposal of hybridoma instability. Hybridoma technology was used to produce anti–*O. viverrini* monoclonal antibody. Previous studies have shown that antibody production via antibody phage display can sustain the hybridoma technique. This paper presents the utility of antibody phage display technology for producing the phage displayed KKU505 Fab fragment and using experiments in concomitant with molecular simulation for characterization. The phage displayed KKU505 Fab fragment and characterization were successfully carried out. The KKU505 hybridoma cell line producing anti–*O. viverrini* antibody predicted to bind to myosin was used to synthesize cDNA so as to amplify the heavy chain and the light chain sequences. The KKU505 displayed phage was constructed and characterized by a molecular modeling in which the KKU505 Fab fragment and -*O. viverrini* myosin head were docked computationally and it is assumed that the Fab fragment was specific to -*O. viverrini* on the basis of mass spectrometry and Western blot. This complex interaction was confirmed by molecular simulation. Furthermore, the KKU505 displayed phage was validated using indirect enzyme-linked immunosorbent assays (ELISA) and immunohistochemistry. It is worthy to note that ELISA and immunohistochemistry results confirmed that the Fab fragment was specific to the -*O. viverrini* antigen. Results indicated that the approach presented herein can generate anti–*O. viverrini* antibody via the phage display technology. This study integrates the use of phage display technology together with molecular simulation for further development of monoclonal antibody production. Furthermore, the presented work has profound implications for antibody production, particularly by solving the problem of hybridoma stability issues.

**Funding:** This research has been funded by the MSc scholarship from the centre for research and development of medical diagnostic laboratories, Liver fluke and cholangiocarcinoma research center, and the Graduate School, Khon Kaen University, Thailand. CN is grateful for financial support by the Center of Excellence on Medical Biotechnology (CEMB), the S&T Postgraduate Education and Research Development Office (PERDO) and the Office of Higher Education Commission (OHEC), Thailand. The funders had no role in study design, data collection and analysis, decision to publish, or preparation of the manuscript.

**Competing interests:** The authors have declared that no competing interests exist.

## Introduction

Cholangiocarcinomas (CCA) are malignancies of the biliary duct system [1]. A considerable amount of literature has been observed on CCA. In 1971–2009, the northeast region of Thailand has witnessed the most incidence of cholangiocarcinomas in which 85 from 100,000 are affected [2–4]. Especially, males in the Khon Kaen province were found to be more prevalent than females with an annual incidence of 84.6 and 36.8 per 100,000, respectively [5]. *Opisthorchis viverrini* (OV) is a major public health problem and is the major association with CCA [1]. Research into OV infection has a long history in Thailand, OV infection was first reported approximately 100 years ago in human. The total number of patients with OV infection in Thailand has increased to be more than 6 million cases, which represents the highest in the world [6]. OV is the most widely distributed parasite in the northeast region of Thailand. Particularly, this parasitic infection had exhibited high prevalence in Thailand for 30 years in which the prevalence in 2009 was 16.6%, which were found to be similar to that of previous report published in 2000 to be 15.7% [7]. These studies illustrate that OV infection is the most serious and widespread disturbances in Thailand. Moreover, the prognosis of OV infection is generally poor whereby most patients are diagnosed at an advanced stage with CCA in which there are no effective treatments available [8,9]. Therefore, the issue of OV infection has received considerable critical attention as is a major public health concern in the northeast region of Thailand and therefore calls for improvement in the diagnostic performance of OV infection.

The gold standard method for the detection of OV infection is the formalin ethyl-acetate concentration technique (FECT), which can quantify OV eggs in feces [9,10]. Despite its efficacy, FECT suffers from several major drawbacks. The disadvantage of this technique is that it is sensitive for only medium to heavy OV infection. Furthermore, the technique lacks sensitivity in the detection of light OV as there is limited analytical specificity by OV eggs that is often confused with minute intestinal flukes infection (MIFs) [11,12]. Moreover, bile duct in advanced stage of CCA can obstruct the flow of eggs into feces thereby making the detection of light OV infection in feces difficult [13,14]. The study of Worasith et al. [15] reported diagnostic methods for the detection of OV infection in humans. It has been suggested that OV antigens in urine that is pre-treated with TCA was detectable by the monoclonal antibody-based enzyme-linked immunosorbent assay (ELISA) as it could measure OV excretory-secretory (OV-ES) antigens (urine OV-ES assay). The urine OV-ES assay was able to determine 28 samples to be positive from a total of 63 samples (44.4%) that is previously detected to be negative by means of the gold standard. This result thereby increases the likelihood of a positive diagnosis of OV infection. The sensitivity and specificity of this method is 81% and 70%, respectively, when compared to the gold standard, FECT. In addition, the use of urine for detecting antigen of the parasites is less invasive and it is easier to handle sample matrices [15]. Therefore, it has been conclusively shown that the urine OV-ES assay serves as a promising alternative diagnosis of patients with OV infection.

The production of monoclonal antibodies is one of the steps in the urine OV-ES assay. In the study by Worasith et al. [15], spleen cells from mice that were previously immunized with the OV-ES antigen were used for fusion with the mouse myeloma cell line for producing the hybridoma cell line. Despite its production success, hybridoma technology has boast a number of problems for their usage. The main technical disadvantage of the hybridoma technology is that the difficulty in processing and the issue of hybridoma instability [16,17]. Hybridoma instability issues arises from the fact that there is instability of each lot of cDNA production due to mutations, chromosome losses, the potential effects of process variables on the yield, quality, homogeneity of the monoclonal antibody (mAb) product as well as the short half-life

to produce the antibody [18,19]. Due to advances in recombinant DNA technology, antibodies can be engineered by means of combinational approaches making use of antibody phage display libraries, which have been reported to be a strong alternative to the hybridoma technology for antibody synthesis [20–23]. Antibody phage display technology is fast becoming a key instrument as it is more efficient than conventional hybridoma system owing to the fact that it is easier to maintain and grow bacterial cultures for recombinant antibody production. Furthermore, there is excellent potential in further improving the binding properties of selected antibody by means of protein engineering techniques [23–25].

This paper proposes a conceptual framework based on the antibody phage display for sustaining monoclonal antibody production, which can play an important role in addressing the aforementioned limitations of the hybridoma technology. Aside from the utilization of phage display technology in the production and characterization of anti-OV antibody on phage particles, this study also makes use of molecular simulation for investigating the antigen-antibody interaction via molecular docking followed by the analysis of their complexation properties by means of molecular simulation. Molecular simulation has been widely shown to be robust and complementary tools in gaining a better understanding of the underlying mechanisms of various biological research questions [25–28]. In this respect, the production of phagemid harboring the sequence of anti-OV antibody was applied for tackling the hybridoma stability issues while the characterization of anti-OV antibody was performed by means of experiments in concomitant with molecular simulation.

## Materials and methods

### *Escherichia coli* and vectors

*E. coli* strains, XL-1 Blue (recA1, endA1, gyrA96, thi-1, hsdR17, supE44, relA1, lac[F' proAB+, LacIq, ZdelM15, Tn10]) (Stratagene, USA) and TG-1 (K-12 supE thi-1 Δ(lac-proAB) Δ(mcrB-hsdSM)5, (rK-mK-)), which was kindly provided by Achara Phumyen, CICM, Thammasat University, Bangkok, Thailand. The pGEM®- T easy vector (Promega, USA) will be used for the pGEM®- T easy vector (promega) system and the pComb3HSS phagemid vector, was a gift from Carlos Barbas, was used for the pComb3HSS system.

### Protein modeling

Amino acid sequences of the *O. viverrini* myosin head [Thanan R] et al. [Unpublished] and the KKU505 Fab fragment were translated from the nucleic acid sequence using the translate tool (http://web.expasy.org/translate/) [29]. Such protein sequences were then used for predicting the protein structure using Zhang's Iterative Threading Assembly Refinement (I-TASSER) server (http://zhanglab.ccmb.med.umich.edu/I-TASSER/) [30–32], which had previously been shown to afford the best protein structure predictions in the Critical Assessment of Structure Prediction (CASP 7 and CASP 8). Best models of the predicted models were selected on the basis of the following parameters: prediction of secondary structure, top five full-length models by the best confidence scores, the estimated TM-score, RMSD, that I-TASSER uses to estimate the quality of the predicted model. The selected model was visualized using the PyMOL program (PyMOL Molecular Graphics System, Version 1.8 Schrödinger, LLC).

### Antibody-antigen docking

Firstly, the protein models of KKU505 Fab fragment and OV Myosin head were predicted using the I-TASSER server [30–32]. Subsequently, docking was applied to predict the protein complex formed by the antibody-antigen interaction using the ClusPro 2.0 server (https://

clusprobu.edu/). ClusPro 2.0 server is a first fully automated web-based program for docking proteins and is one of the top performers at Critical Assessment of Predicted Interactions (CAPRI) [33–36]. The obtained antibody-antigen complex results from the server were ranked and the best antibody-antigen complex that represents the best match with results from mass spectrometry was selected. The selected antibody-antigen complex was analyzed using the PyMOL program. In addition, a 2D illustration of the complex was visualized by the LIGPLOT tool [37].

## Production of phage display KKU505 Fab fragment

A single clone of *E. coli* TG-1 harboring pComb3HSS phagemid vector with KKU505 Fab fragment gene (submitted paper) was chosen from a 2x YT agar plate containing ampicillin. In brief, transformed bacteria were grown in 2x TY broth containing 50 μg/ml ampicillin at 37˚C with shaking at 200 rpm. The precultured bacteria were subsequently transferred to the same medium and culture at 37˚C with shaking at 200 rpm until the optical density at 600 nm (OD600) reach 0.8. After induction, the bacterial culture was further infected with $10^{12}$ pfu/ml of VCSM13 helper phages and left at 37˚C for 2 hours with shaking, before 70 μg/ml kanamycin was added to the culture and incubated at 37˚C for 18 hours on a shaker. The cell of the culture was spin down and transferred supernatant to a clean bottle. The supernatant was added 5% (w/v) of PEG-8000 and 3% (w/v) of NaCl, before placed on a shaker for 5 minutes to mix the reagent and precipitated phage on ice for 30 minutes. Following, spin down precipitate at 4˚C for 20 minutes, discarded supernatant to remove PEG solution as much as possible. Finally, the pellet was suspended in TBS/1% BSA and stored the supernatant at 4˚C.

## Characterization of phage display of KKU505 Fab fragment by indirect ELISA

Microtiter plates were coated with 1 μg/well of crude *O. vivirrini* in carbonate/bicarbonate buffer pH 8.6 overnight at 4˚C. The plates were then blocked with 300 μl of 5% bovine serum albumin (BSA) in PBS for 2 hours at 37˚C. The wells were washed with 0.02% Tween-20 in TBS and 100 μl of $10^{12}$ PFU/ml of phage in PBS were added and incubated for 2 hours 37˚C. The unbound phage was washed out by with 0.01% Tween-20 in TBS for 5 times. HRP-conjugated anti-M13 major coat protein antibody (1:800, Santa Cruz Biotechnology, sc-53004 HRP) was added and incubated for 2 hours at 37˚C. Bound phage was detected by adding 100 μL/well of 3,3_,5,5_-tetramethylbenzidine (TMB) substrate (KPL, Gaithersburg, MD), which containing with substrate A and B at 1:1 (v/v) ratio, after that it was incubated in the dark at room temperature for 30 minutes. After the blue color was developed, in each well was added 50 μL of the stop reaction solution (2N H2PO4). The blue color changed to yellow color and was measured the absorbance (optical density; O.D.) at 450 nm by Sunrise ELISA plate reader (Tecan, Austria) with Magellen7 software (Tecan, Austria).

## Characterization of phage display of KKU505 Fab fragment by immunohistochemistry

Deparaffinized the biliary system of mouse infected with *O. viverrini* section were treated with 0.3% $H_2O_2$ for 30 minutes to digest endogenous peroxidase. After blocking nonspecific staining with 10% skim milk for 30 min. These sections were incubated overnight at 4˚C with $10^{12}$ PFU/ml of phage display KKU505 Fab and KKU505 monoclonal antibody (1:500). After washing, the sections were sequentially incubated for 90 minutes with HRP-conjugated anti-M13 major coat protein antibody (1:1000, Santa Cruz Biotechnology, sc-53004 HRP) for phage

system and HRP-anti-mouse antibody (Envision) for monoclonal system and then developed in diaminobenzidine tetrahydrochloride (DAB). After that, hematoxylin was used to counter-staining the section. These sections were dehydrated, cleared and mounted. KKU505 mAb was used for positive pattern with mouse Envision Systems. The localization of *O. viverrini* antigens was performed using the Immunohistochemistry with HRP-conjugated anti-M13 major coat protein antibody for phage system and diaminobenzidine tetrahydrochloride (DAB) for monoclonal system.

## Results

The utilization of the hybridoma technology in producing KKU505 mAb for OV infection detection possess limitations pertaining to their stability and genetic changes that affects the quality of this mAb. For solving this problem, preservation of the antibody sequence in plasmids is a viable solution for long term stability. Therefore, the aim of this study is to construct the pComb3HSS phagemid vector with the KKU505 Fab sequence followed by experimental investigations and molecular simulations for characterizing the properties of KKU505 Fab fragment. A summary of the conceptual framework associated in this study is provided in Fig 1 and as can be seen this work can be divided into 4 clusters. In a nutshell, Phagemid vector carrying KKU505 Fab sequence construction (blue dashed box), each KKU505 Fab fragment was ligated into the pGEM®-T easy vector before transferring to the pComb3HSS phagemid vector. Phage display KKU505 Fab fragment was successfully produced and is validated through the combined use of indirect phage ELISA and immunohistochemistry experiments (green dashed box). Furthermore, molecular simulation by means of antigen-antibody docking was performed so as to characterize this complex (yellow dashed box) as shown in green and yellow dashed box, respectively. The molecular simulation has performed the docking of KKU505 mAb and OV myosin head (mass spectrometry result) models as antigen-antibody interaction in Western blot results (red dashed box). Results in this study shown successfully constructed pComb3HSS phagemid vector carrying the anti-OV antibody.

### Characterization of phage display KKU505 Fab fragment

For validation of phage display KKU505 Fab fragment, C.HM.LC. 1 and 8 (submitted paper) was used to rescues phage particles display KKU505 Fab fragment names KKU505 phage 1 and 8 respectively.

**Characterization using indirect ELISA.** For characterizing the specificity of antigen-specific binding of phage presenting KKU505 Fab fragment (KKU505 phage 1 and 8), VCSM13 helper phage (i.e., a phage prepared from non-transformed TG-1 and phage display 1H10 antibody, which is a phage against the MICA antigen) did not generate the signal against OV as antigens were verified by indirect ELISA for the detection of crude OV antigens (Fig 2). The ELISA signal of phage display KKU505 Fab fragment against crude OV antigens were found to be significantly higher than those of VSCM and 1H10 Phage (P-*value* <0.0001). This result suggests that phage display KKU505 Fab fragment was specific to crude OV antigens and had been successfully produced.

As per the sensitivity or concentration limitation, KKU505 phage 8 was characterized for their ability to determine relation between crude OV antigens concentration and KKU505 phage 8 using indirect ELISA. Phage display KKU505 Fab fragment demonstrated relation between ELISA signal and OV antigens at different range of concentration as compared with that of bovine serum albumin (BSA) encompassing 0, 0.05, 0.1, 0.25, 0.5, 1 and 2 μg/well. The signal from the lowest OV antigens concentration (0.05 μg/well) was still significantly higher than that of all BSA concentration (P-*value* <0.0001). Additionally, this result shows the

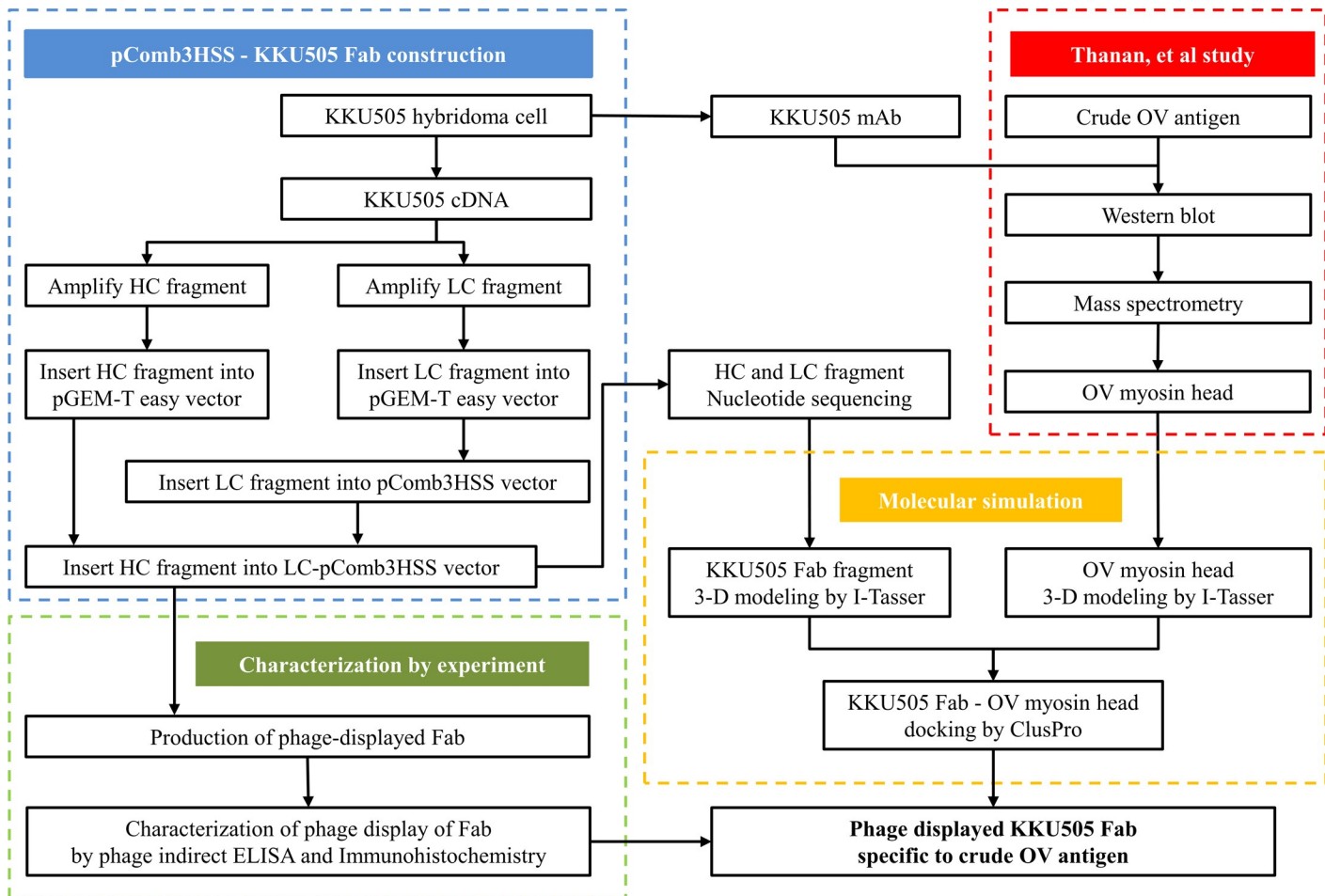

**Fig 1. Conceptual framework for this study.** This framework can be divided to 4 major cluster; Construction of phagemid vector carrying KKU505 Fab sequence (blue dashed box), characterization of KKU505 Fab fragment by experiment (green dashed box) and molecular simulation (yellow dashed box), and Thanan et al. study (red dashed box).

relation between ELISA signal and OV antigens concentration. Furthermore, the results additionally suggested that the phage display KKU505 Fab fragment was specific to crude OV antigens and that KKU505 phage 8 can detect OV antigens at a concentration of 0.05 μg/well using indirect ELISA (Fig 3).

**Characterization using immunohistochemistry.** Localization and reaction pattern of phage presenting KKU505 Fab fragment (KKU505 phage 8) was verified by immunohistochemistry on the biliary system of mouse infected with *O. viverrini* (Fig 4). The localization and reaction of OV antigens on the biliary system of mouse infected with *O. viverrini* were detected by KKU505 monoclonal antibody as positive reaction for OV antigens and PBS as negative reaction for OV antigens. Phage presenting KKU505 Fab fragment shows strongly positive localization and reaction on the biliary system of mouse infected with *O. viverrini* when compared with that of the KKU505 monoclonal antibody. Results suggested that the phage display KKU505 Fab fragment was specific to OV antigens and had been successfully produced and shown to exhibit localization of this phage presenting KKU505 Fab fragment. The reaction or pattern of immunohistochemistry using both phage display KKU505 Fab fragment and KKU505 monoclonal antibody shows strongly positive results in parasite, biliary epithelium and surrounding tissues.

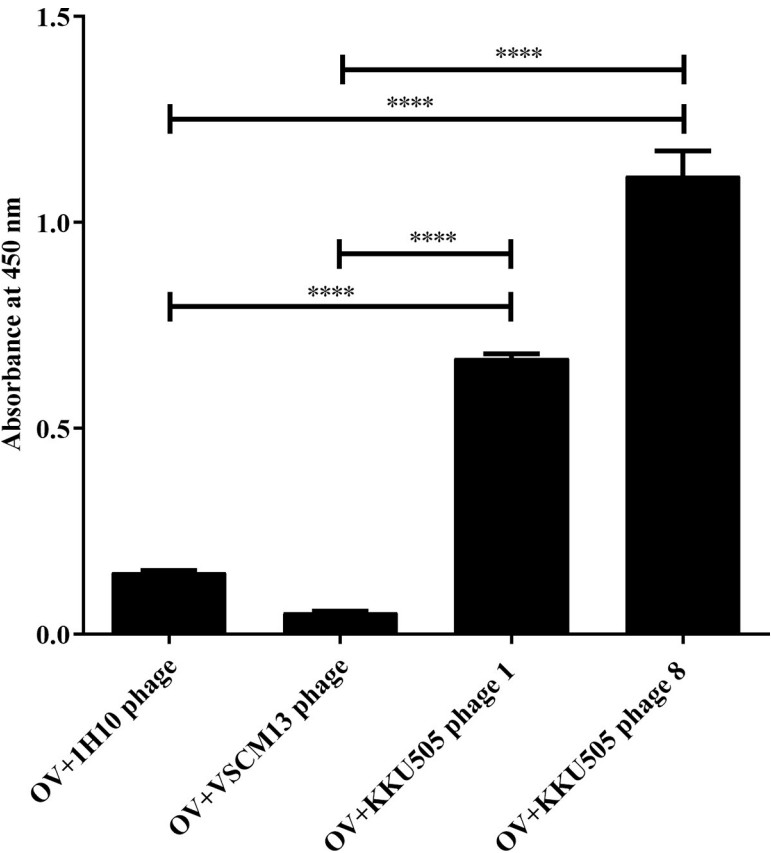

**Fig 2. Validation of phage display KKU505 Fab fragment by indirect ELISA.** Crude OV antigens at concentration of 1 μg/well were captured on wells. $10^{12}$ pfu/well of KKU505 Fab phage 1 and 8, 1H10 phage (phage display anti-MICA Fab fragment) and VSCM13 phage (helper phage) was performed using indirect ELISA, and traced by HRP-conjugated anti-M13 major coat protein antibody and 1H10 phage was used as a negative control. This phage ELISA signals ware normalized by reaction signal without phage (mean OD is 0.199).

## KKU505 antibody properties

After nucleotide sequencing, KKU505 heavy chain Fd and light chain sequences of C.HM. LC.8 was subjected to analysis. The KKU505 heavy chain Fd and light chain sequences were identified via the international ImMunoGeneTics information system (IMGT) [38–40] to assure success in amplification and cloning of the immunoglobulin Fab fragment. Particularly, this KKU505 heavy chain Fd and light chain sequences were generated via 3D protein structure prediction using I-TASSER [30–32] and subsequently analyzed using the PyMOL program (The PyMOL Molecular Graphics System, Version 1.8, Schrödinger, LLC.). In regards to the KUU505 Fab 3D model, this structure resembles an arm in the Y-shape conformation, which can exist in antibody structure for epitope recognition (Fig 5A). Electrostatic potential surfaces of this KKU505 Fab protein structure model were calculated using the APBS Tool 2.1 (plugin integrates the APBS software package into PyMOL) [41]. The Epitope binding site of this model consists mostly of positive charge residues (Fig 5B). For the KKU505 heavy chain Fd and light chain nucleotide sequence, these sequences, which were also identified by applying an antibody numbering scheme were analyzed for CDR loops using the IMGT [40] in which all were suggested to have CDRs (Fig 5C).

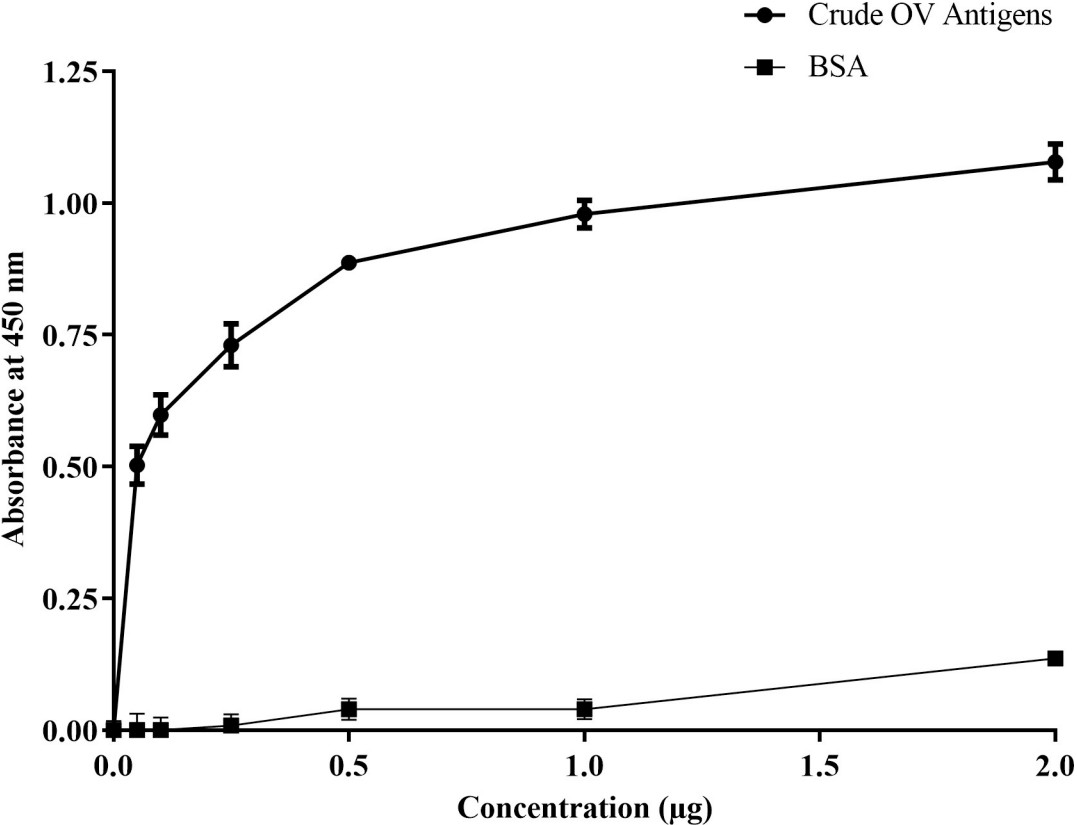

**Fig 3. Relation curve between crude OV antigens and bovine serum albumin (BSA) concentration and phage detection by indirect ELISA.** A standard curve was plotted with crude OV antigens bovine serum albumin (BSA) with concentrations ranging from 0.01 μg/well to 2 μg/well. $10^{12}$ pfu/well of KKU505 Fab phage 8 was used as a detector using indirect ELISA technique. This phage ELISA signals ware normalized by reaction signal without antigens (mean OD is 0.224).

### KKU505 Fab -*O. viverrini* myosin head docking

In a study conducted by Thana et al., mass spectrometry was used (submitted paper) for the identification of crude OV antigens that is specific to the KKU505 monoclonal antibody. It was found that the antigen was an OV myosin head (accession number is OON15278.1). The amino acid sequence of the OV myosin head was used as input for generating the 3D structure model by means of I-TASSER [30–32]. The 3D models of both KKU505 Fab and OV myosin head were validated by analyzing the Ramachandran plot, which was performed using the PyMod plugin in PyMOL and was found to be energetically in the allowed regions. Most residues of OV Fab and the myosin head were found to be in the favorable region of the protein as modeled by the I-TASSER server (S1 Fig). KKU505 Fab and OV myosin head models were subjected to protein-protein docking using the ClusPro 2.0 server [33–36]. The best complex that best matches with mass spectrometry result was selected and analyzed using PyMOL program to see this antigen-antibody complex interaction (Figs 6 and S2).

## Discussion

We describe the creation of an anti-*O. viverrini* Fab antibody phage display and their characterization. An initial objective of the project was the production of anti-*O. viverrini* antibody, which can be used for the diagnostic application particularly in *O. viverrini* infection detection in urine samples. In the work of Worasith [15], the anti-*O. viverrini* antibody was produced

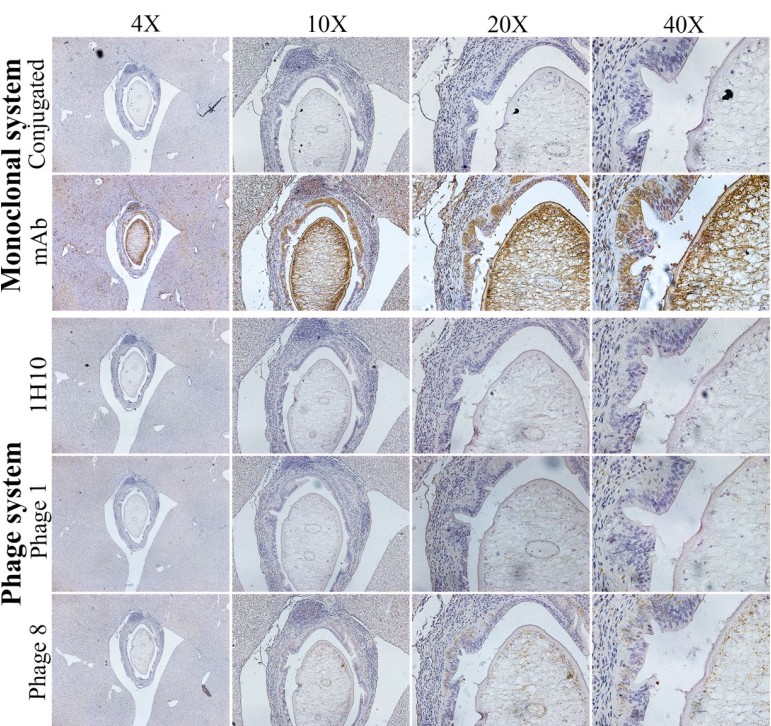

**Fig 4. Immunohistochemistry staining on the biliary system of mouse infected with -*O. viverrini*.** Comparison of Immunohistochemistry staining on the biliary system of mouse infected with -*O. viverrini* which detected by PBS and KKU505 monoclonal antibody as a monoclonal system and KKU505 phage 1 and 8 and 1H10 phage (phage display anti-MICA Fab fragment) as a phage system. PBS and 1H10 phage as negative control of monoclonal system and phage system, respectively. KKU505 phage 1 and 8 for localization pattern comparison of phage display KKU505 Fab fragment specific to OV antigens and KKU505 mAb. "Conjugated" is conjugated control. "mAb" is KKU505 mAb. "1H10" is phage display anti-1H10 antibody. "Phage 1 and 8" is phage display antibody KKU508 clone 1 and 8, respectively.

using the KKU505 hybridoma cell line produced by fusing mouse myeloma cell line (clone P3x63-Ag8.653) with spleen cells from BALB/c mice that is previously immunized with the OV-ES antigen [15]. The weak association of anti-*O. viverrini* antibody production is hybridoma instability. Several reports have reported that the hybridoma technique had many limitations. Particularly, there are no practical ways to alter the properties or improve antibody that are produced by hybridoma and they are often associated with mutations, chromosome losses, time-consuming, expensive and short halt-life [16,17]. Phage display has been introduced as an efficient alternative to hybridoma technology [20–23], this study aims for antibody production using the phage display technology because this technique is capable of constructing phagemid harboring antibody sequence. For improved half-life, antibody sequence preservation in phagemid is preferable than preservation in hybridoma cells because nucleic acids are more stable than cell lines [42].

The expression of antibody on the surface of phage particles represents a powerful approach for the isolation of monoclonal antibody with bio-panning that can be used in the design and construction of antibody fragments [43,44], therefore antibody phage display can be applied to improve the affinity of antibodies using the mutagenesis technique while also affording easy isolation of a single clone from the mutagenesis technique so as to select the optimal phage display antibody. Several experiments corroborated this idea where antibody against the major histocompatibility complex class I [45] and human monoclonal antibody against the third

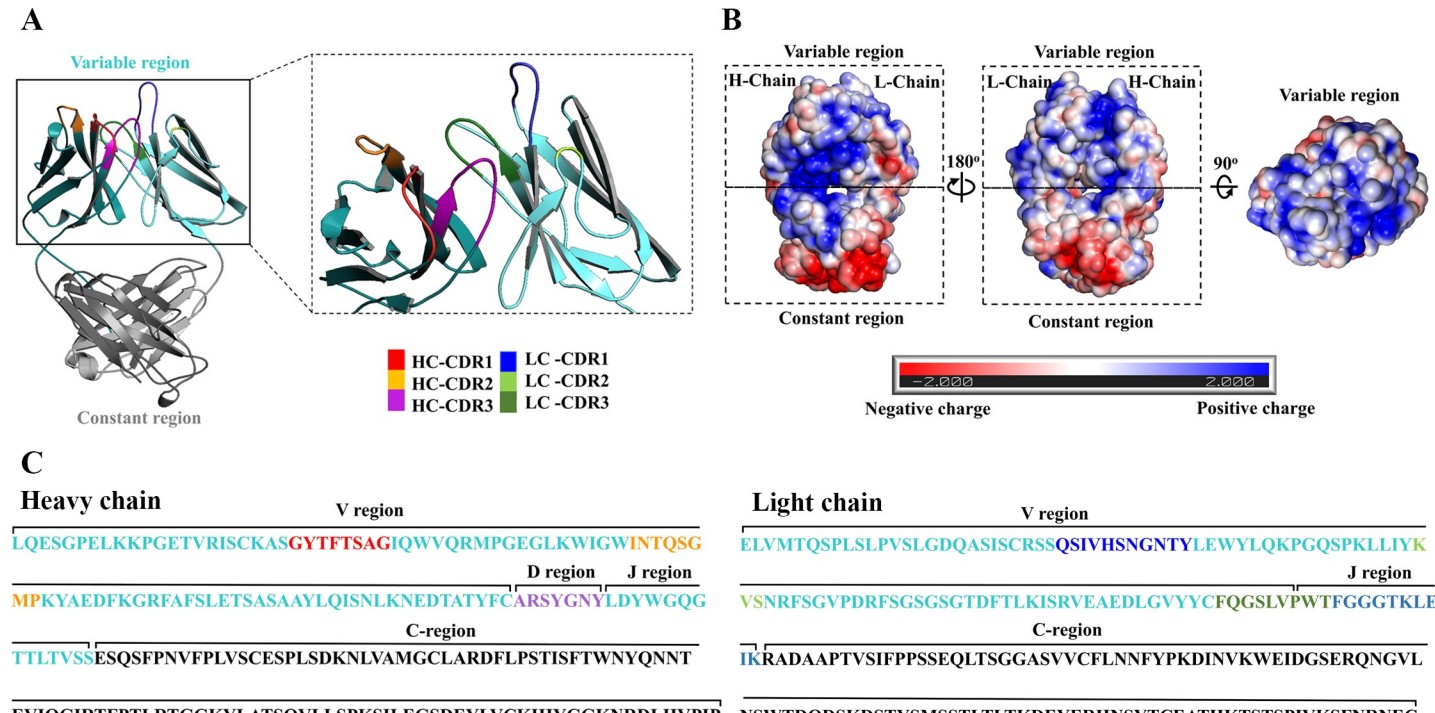

**Fig 5. Structure and sequence of KKU505 Fab modelling.** A. The 3D structure of KKU505 Fab was predicted using the I-TASSER server. This structure was colored following IMGT (cyan: Variable region and gray: Constant region). B. Electrostatic potential of the 3D structure of KKU505 Fab colored according to charge (blue: Positive charge and red: Negative charge). C. The sequence of KKU505 Fab was colored following IMGT (cyan: Variable region, gray: Constant region, red: HC-CDR1, orange: HC-CDR2, purple: HC-CDR3, blue: LC-CDR1, green: LC-CDR2 and forest: LC-CDR3). "HC" is heavy chain. "LC" is light chain. "CDR" is complementarity determining region.

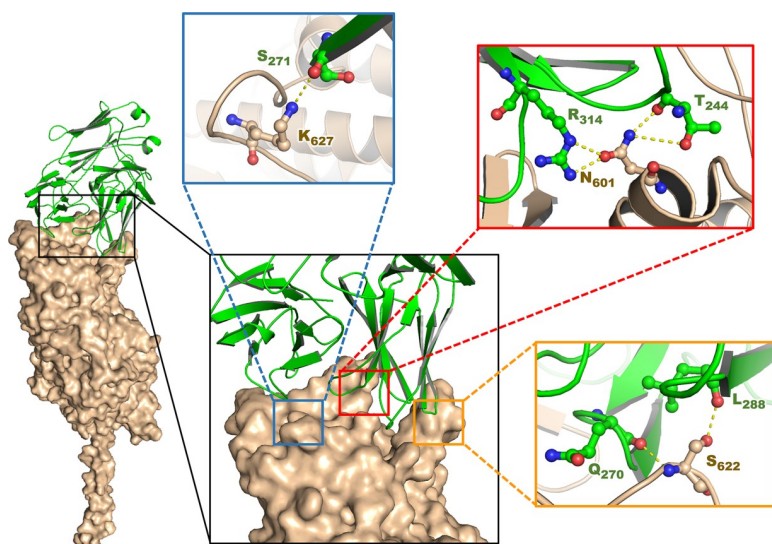

**Fig 6. Antibody–antigen docking output.** Molecular interactions between KKU505 Fab (green schematic ribbon) and OV myosin head (wheat surface) as antibody–antigen complex output by ClusPro 2.0. Zoomed regions show interactions of this antibody–antigen complex. Side-chains are ball and stick. Hydrogen and ionic bonds are dotted lines. Amino acids are abbreviated with a capital. The number is residue position.

hypervariable loop of human immunodeficiency virus were shown to afford improved binding activity as evaluated by phage display technology [46]. This finding broadly supports the anti-*O. viverrini* antibody affinity improvement that can consequently be used to increase OV detection in urine so as to afford better diagnosis with more sensitivity and specificity in the future.

The characterization and validation of phage display Fab fragment by means of experiments in concomitant with molecular simulation. Results of indirect ELISA and antibody-antigen docking demonstrated that this KKU505 Fab fragment as displayed on the phage particle was specific to the OV antigen. Specifically, antibody-antigen docking was performed using prior mass spectrometry results (from the KKU505 mAb as well as crude OV antigens interaction using Western-blot) to drive the docking simulations so as to gain a better understanding of the underlying mechanisms of this interaction.

As for antibody modeling, Fernández-Quintero and co-workers [47] suggested the importance of Fab structure and special care should be taken into consideration. Particularly, the antibody crystal structure without the antigen present was found to bind to the tail region of the Fab. In this study, the modeled Fab structure in the absence of the antigen was used whereby the best antibody-antigen complex model was selected by avoiding antigen binding to the tail of the Fab structure. The linkage between the KKU505 Fab fragment and OV antigen was investigated using molecular simulation. Particularly, molecular dynamics is instrumental for rationalizing the protein-protein interaction and its conformation over the simulation. This was calculated using the YASARA software package [48] with the AMBER14 force field for the duration of 15 nanoseconds timescale (unpublished data). Findings from molecular dynamics simulation conducted by Fernández-Quintero et al. [49] shows that the CDR loop ensemble of antibodies exert its effects on the paratope states in solution. Furthermore, it was found that CDR loops can shift the relative VH and VL orientation, the elbow-angle structure distributions, and the dynamic simulation [50]. Thus, these aforementioned results show that the antibody CDR loops plays an important role in the antibody properties. These results suggested that the antibody CDR loops is an important region that should be considered in antibody structure design as well as in molecular simulation. However, the analysis of protein-protein interaction as investigated through molecular simulation may needs more analysis timescale.

The combination of molecular simulation and experiment represents a powerful approach for dissecting the antigen-antibody interaction. Experiments conducted by Lousa et al. [51] shows that one of the major causes of lowered activity of mutants of the influenza fusion peptide is due to their lowered membrane affinity as deduced from the combination of simulation and experimental techniques. Moreover, Pomés et al. [52] also demonstrated the benefits of computational methods in conjunction with experiments for studying allergen-antibody interaction and allergens antibody binding epitopes whereby antibody phage display technology was used in combination with a computational algorithm for identifying the allergen surface that mimics conformational epitopes. As for the experiment, immunohistochemistry was used to demonstrate the localization of the OV antigen. The study of Sripa and Kaewkes [53] demonstrated sequential parasite antigen localisation in the biliary system of hamsters infected with *O. viverrini* detected by Immunohistochemistry. The reaction was performed using antibody to bind somatic or ES antigens. Positive staining revealed that parasite antigens are seen within the adult *O. viverrinias* and can be gradually observed in the mucus coat, biliary epithelium and surrounding tissue [53]. The pattern on the biliary system of hamsters infected with *O. viverrini* was detected by anti-ES antibody. This was found to be similar to the pattern of the biliary system of mouse infected with *O. viverrini* as detected by phage display KKU505 Fab. As such, the phage displayed KKU505 Fab is specific to ES antigens of *O. viverrini*.

For further experiments, the anti-*O. viverrini* fragment as displayed on the phage particle can be used to confirm the interaction with the recombinant OV myosin head. This can adequately validate and characterize the antibody properties primarily by using Western blot for determining the molecular weight of specific antigen, indirect ELISA for determining the specificity of this antibody toward antigens of interest, Flow cytometry for validating the interaction and affinity of this antibody and bio-layer interferometry for measuring the affinity of this antibody. Phage display anti-*O. viverrini* fragment or soluble anti-*O. viverrini* fragment should then be applied for detecting OV-ES antigens in urine sample while ELISA is used for determining the efficiency of OV infection diagnosis ability of them.

## Conclusion

In summary, we have demonstrated the effective combination of experiment and molecular simulation for validating the interaction of the KKU505 Fab fragment and the OV antigen. The aim of this study was to construct phagemid carrying KKU505 Fab sequence that can be used to produce the phage displayed KKU505 Fab fragment for antibody production. Subsequently, the KKU505 Fab fragment was characterized using experimental and bioinformatics tools. Firstly, phage indirect ELISA was shown to the KKU505 Fab fragment that demonstrated specificity towards the crude OV antigens. Such findings suggest that the phage display technology can be used to produce and characterize KKU505 Fab fragments on phage particles. Secondly, the antigen-antibody interaction of OV antigen-KKU505 Fab fragment was elucidated via molecular docking. Moreover, this interaction was also investigated experimentally via Western blot by staining crude OV antigens and KKU505 Fab mAb. All results indicated that the KKU505 Fab fragment, which is displayed on the phage particle, was specific to the crude OV antigens. Such findings presented herein demonstrates the high potential of the phage display technology for monoclonal antibody production.

## Supporting information

**S1 Fig. Validation of modelled structure by Ramachandran plot using PyMod 2.0.** KKU505 Fab and OV myosin head structure structures were validated by Ramachandran plot (red: Favoured region, yellow: Allowed regions, pale yellow: Generously allowed regions and write: Disallow regions).
(TIF)

**S2 Fig. All molecular interactions between antibody–antigen complexes by docking.** Interactions between KKU505 Fab (green schematic ribbon) and OV myosin head (wheat surface) by ClusPro 2.0. Side-chains are ball and stick. Hydrogen and ionic bonds are dotted lines. Amino acids are abbreviated with a capital. The number is residue position.
(TIF)

**S1 File. Minimal data set.**
(DOCX)

## Acknowledgments

The authors are grateful to A. Phumyen is acknowledged for the generous gift of *E. coli* strains, TG-1, and the phage display technology assistance. We are also grateful to Prof. Paiboon Sithithaworn and Mrs. Chanika Worasith for supporting KKU505 hybridomas cell line, Mrs. Rungtiwa Nutalai for PCR amplification and cloning assistances. Mr. Phongsaran Kimawaha and Mr. Achira Namjan for Immunohistochemistry assistance. Dr. Aijaz Ahmad Malik and

Prof. Dr. Chanin Nantasenamat, Center of Data Mining and Biomedical Informatics, Faculty of Medical Technology, Mahidol University for their advices and molecular protein modeling and docking assistance.

## Author Contributions

**Conceptualization:** Sitthinon Siripanthong, Anchalee Techasen, Paiboon Sithithaworn, Chanvit Leelayuwat, Amonrat Jumnainsong.

**Data curation:** Sitthinon Siripanthong, Chanvit Leelayuwat, Amonrat Jumnainsong.

**Formal analysis:** Sitthinon Siripanthong, Anchalee Techasen, Chanin Nantasenamat, Aijaz Ahmad Malik, Chanvit Leelayuwat, Amonrat Jumnainsong.

**Funding acquisition:** Amonrat Jumnainsong.

**Investigation:** Chanvit Leelayuwat, Amonrat Jumnainsong.

**Methodology:** Sitthinon Siripanthong, Anchalee Techasen, Chanin Nantasenamat, Aijaz Ahmad Malik, Chanvit Leelayuwat, Amonrat Jumnainsong.

**Project administration:** Sitthinon Siripanthong, Anchalee Techasen, Amonrat Jumnainsong.

**Resources:** Anchalee Techasen, Chanin Nantasenamat, Aijaz Ahmad Malik, Chanvit Leelayuwat, Amonrat Jumnainsong.

**Software:** Chanin Nantasenamat, Aijaz Ahmad Malik.

**Supervision:** Anchalee Techasen, Chanin Nantasenamat, Aijaz Ahmad Malik, Paiboon Sithithaworn, Chanvit Leelayuwat, Amonrat Jumnainsong.

**Validation:** Anchalee Techasen, Chanin Nantasenamat, Aijaz Ahmad Malik, Paiboon Sithithaworn, Chanvit Leelayuwat.

**Visualization:** Chanin Nantasenamat, Aijaz Ahmad Malik, Paiboon Sithithaworn, Chanvit Leelayuwat.

**Writing – original draft:** Sitthinon Siripanthong, Amonrat Jumnainsong.

**Writing – review & editing:** Anchalee Techasen, Chanin Nantasenamat, Aijaz Ahmad Malik, Paiboon Sithithaworn, Chanvit Leelayuwat.

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
