## [Decision Letter · Decision Letter 0]

13 Nov 2020

PONE-D-20-29938

Production and characterizationof antibody against Opisthorchis viverrini via phage display and molecular simulation

PLOS ONE

Dear Dr. Jumnainsong,

Thank you for submitting your manuscript to PLOS ONE. After careful consideration, we feel that it has merit but does not fully meet PLOS ONE’s publication criteria as it currently stands. Therefore, we invite you to submit a revised version of the manuscript that addresses the points raised during the review process.

We look forward to receiving your revised manuscript.

Kind regards,

Theerapong Krajaejun, M.D.

Academic Editor

PLOS ONE

Additional Editor Comments:

The manuscript entitled " Production and characterization of antibody against Opisthorchis viverrini via phage display and molecular simulation" was reviewed by an expert in the field. The manuscript has now been returned to the authors for major revision. There are some issues relating to technical/statistical analyses and data availability to support the conclusion of the study that need to address (see the attached).

Journal Requirements:

2. Please describe the source of the KKU505 monoclonal antibody, and ensure that enough details are provided so that another researcher could generate or obtain this antibody.

Also, please include all antibody dilutions for all experiments.

3. We note that you have reference to data in the manuscript which has currently not yet been accepted for publication:  'Amino acid sequences of the -O. viverrini myosin head (accession number is OON15278.1) (i.e. identified from from mass spectrometry; unpublished results)..'      

Please remove this from your References and amend this to state in the body of your manuscript: (ie “[XX Name] et al. [Unpublished]”) as detailed online in our guide for authors: http://journals.plos.org/plosone/s/submission-guidelines#loc-reference-style

"We also gratefully acknowledge Dr. Carlos F. Barbas III for providing pComb3HSS. CN is grateful for

financial support by the Center of Excellence on Medical Biotechnology (CEMB), the

S&T Postgraduate Education and Research Development Office (PERDO) and the

Office of Higher Education Commission (OHEC), Thailand. [13]"

"This research has been funded by the MSc scholarship from the centre for research and development of medical diagnostic laboratories, Liver fluke and cholangiocarcinoma research center, and  the Graduate School, Khon Kaen University, Thailand. The funders had no role in study design, data collection and analysis, decision to publish, or preparation of the manuscript."

Reviewers' comments:

Reviewer's Responses to Questions

**Comments to the Author**

1. Is the manuscript technically sound, and do the data support the conclusions?

Reviewer #1: No

2. Has the statistical analysis been performed appropriately and rigorously? 

Reviewer #1: No

3. Have the authors made all data underlying the findings in their manuscript fully available?

Reviewer #1: No

4. Is the manuscript presented in an intelligible fashion and written in standard English?

Reviewer #1: Yes

5. Review Comments to the Author

Reviewer #1: The paper entitled “Production and characterization of antibody against Opisthorchis viverrine via phage display and molecular dynamics simulations “addresses on the one hand the issue of safe disposal of hybridoma instability and on the other hand shows the utility of phage display and MD simulation to produce and characterize the KKU505 antibody.

The study mentions in the title and in the discussion an MD simulation of 15 ns, but neither in the methods nor in the Supporting Information the parameters used for MD simulations are shown. Were the simulations performed in implicit or explicit solvent, which force field was used, which thermostat, barostat, ...

Apart from that also no MD simulation data is shown (RMSD plot, clustering...). Are 15 ns enough to draw any conclusions, especially when the complex structure was determined by docking, a longer relaxation time of the complex might be necessary. Additionally, the antibody structure was also modelled, which introduces another uncertainty. Recent literature described the timescales of CDR loop rearrangements in combination with also the fast VH-VL interdomain orientations, which also influence the shape of the paratope and might also have an effect on the resulting docked complex. Additionally, please also take the following review about allergen-antibody interactions into account.

https://doi.org/10.1080/19420862.2020.1744328, https://www.ncbi.nlm.nih.gov/pmc/articles/PMC7492603/
https://www.nature.com/articles/s42003-020-01319-z.

Please discuss the timescale problem and also please mention if the template used for the antibody homology model was based on an unbound Fab structure. If yes please also discuss the problems of potential crystal packing effects, which can distort the CDR loops, which are involved in binding. (https://www.tandfonline.com/doi/full/10.1080/19420862.2019.1618676,

Please comment to this.

6. PLOS authors have the option to publish the peer review history of their article (what does this mean?). If published, this will include your full peer review and any attached files.

Reviewer #1: No

---

## [Author Response · Author response to Decision Letter 0]

23 Jan 2021

Response to academic editor and reviewer

PLOS ONE

Article Title: Production and characterization of antibody against Opisthorchis viverrini via phage display and molecular simulation

Running Number: PONE-D-20-29938

Response to academic editor 

https://journals.plos.org/plosone/s/file?id=wjVg/PLOSOne_formatting_sample_main_body.pdf
https://journals.plos.org/plosone/s/file?id=ba62/PLOSOne_formatting_sample_title_authors_affiliations.pdf

 Response: This manuscript entitled “Production and characterization of antibody against Opisthorchis viverrini via phage display and molecular simulation” was changed the file form from Latex file to Word file base on PLOS ONE's style requirements and formatting. 

2. Please describe the source of the KKU505 monoclonal antibody, and ensure that enough details are provided so that another researcher could generate or obtain this antibody.

Also, please include all antibody dilutions for all experiments.

 Response: The source of the KKU505 monoclonal antibody was added in this revised manuscript on page 10, line 315-317 as shown in the red letter. This description is “the KKU505 hybridoma cell line produced by fusing mouse myeloma cell line (clone P3x63-Ag8.653) with spleen cells from BALB/c mice that is previously immunized with the OV-ES antigen [15]” 

The antibody dilutions and phage concentration for all experiments were added in this revised manuscript as shown in the red letter. There are

 on page 5, line 147 “1012 PFU/ml of phage in PBS”

 on page 6, line 149 “1:800”

on page 6, line 163 “1012 PFU/ml of”

on page 6, line 163 “1:500”

on page 6, line 165 “1:1000”

3. We note that you have reference to data in the manuscript which has currently not yet been accepted for publication: 'Amino acid sequences of the -O. viverrini myosin head (accession number is OON15278.1) (i.e. identified from from mass spectrometry; unpublished results).' 

Please remove this from your References and amend this to state in the body of your manuscript: (i.e. “[XX Name] et al. [Unpublished]”) as detailed online in our guide for authors: http://journals.plos.org/plosone/s/submission-guidelines#loc-reference-style.

 Response: The reference “[Thanan R] et al. [Unpublished]” was instead of “(accession number is OON15278.1) (i.e. identified from from mass spectrometry; unpublished results)” in this revised manuscript as shown in the red letter on page 4, line 102-103.

"We also gratefully acknowledge Dr. Carlos F. Barbas III for providing pComb3HSS. CN is grateful for financial support by the Center of Excellence on Medical Biotechnology (CEMB), the S&T Postgraduate Education and Research Development Office (PERDO) and the

Office of Higher Education Commission (OHEC), Thailand."

"This research has been funded by the MSc scholarship from the centre for research and development of medical diagnostic laboratories, Liver fluke and cholangiocarcinoma research center, and the Graduate School, Khon Kaen University, Thailand. The funders had no role in study design, data collection and analysis, decision to publish, or preparation of the manuscript."

 Response: The funding-related text “SS is holding an MSc scholarship from the centre for research and development of medical diagnostic laboratories, Khon Kaen University, Khon Kaen, Thailand and liver fluke and cholangiocarcinoma research center, faculty of medicine, Khon Kaen University, Khon Kaen, Thailand, the Graduate School Khon Kaen University, Thailand.” was removed from this revised manuscript as shown in the red letter on page 13, line 410-413.

 The funding-related text “We also gratefully acknowledge Dr. Carlos F. Barbas III for providing pComb3HSS. CN is grateful for financial support by the Center of Excellence on Medical Biotechnology (CEMB), the S&T Postgraduate Education and Research Development Office (PERDO) and the Office of Higher Education Commission (OHEC), Thailand.” was removed from this revised manuscript as shown in the red letter on page 13, line 420-424.

“We are also grateful to Prof. Paiboon Sithithaworn and Mrs. ChaniKa Worasith for supporting KKU505 hybridomas cell line, Mrs. Rungtiwa Nutalai for PCR amplification and cloning assistances. Mr. Phongsaran Kimawaha and Mr. Achira Namjan for Immunohistochemistry assistance. Dr. Aijaz Ahmad Malik and Prof. Dr. Chanin Nantasenamat, Center of Data Mining and Biomedical Informatics, Faculty of Medical Technology, Mahidol University for their advices and molecular protein modeling and docking assistance.” Was added to this revised manuscript as shown in the red letter on page 13, line 415-420.

 The Funding Statement no need to update. You can publish funding information present in the Funding Statement section.

Response to reviewer #1

1. Is the manuscript technically sound, and do the data support the conclusions?

Reviewer #1: No

2. Has the statistical analysis been performed appropriately and rigorously?

Reviewer #1: No

 Response: All statistics (an unpaired t-test) performed in Fig 2. (Validation of phage display KKU505 Fab fragment by indirect ELISA) were calculated the power of this test using G*Power 3.1.9.7. The method, input, and output of these analyses will be shown in the following paragraphs. There are show 100% Power (1-β err prob).

KKU505 Phage 1 VS 1H10 MICA Phage

t tests - Means: Difference between two independent means (two groups)

Analysis: Post hoc: Compute achieved power 

Input: Tail(s) = Two

 Effect size d = 15.72872

 α err prob = 0.05

 Sample size group 1 = 3

 Sample size group 2 = 3

Output: Noncentrality parameter δ = 19.2636692

 Critical t = 2.7764451

 Df = 4

 Power (1-β err prob) = 1.0000000 (100%)

KKU505 Phage 1 VS VSCM13 helper phage

t tests - Means: Difference between two independent means (two groups)

Analysis: Post hoc: Compute achieved power 

Input: Tail(s) = Two

 Effect size d = 16.46065

 α err prob = 0.05

 Sample size group 1 = 3

 Sample size group 2 = 3

Output: Noncentrality parameter δ = 20.1600967

 Critical t = 2.7764451

 Df = 4

 Power (1-β err prob) = 1.0000000 (100%)

KKU505 Phage 8 VS 1H10 MICA Phage

t tests - Means: Difference between two independent means (two groups)

Analysis: Post hoc: Compute achieved power 

Input: Tail(s) = Two

 Effect size d = 20 (25.23094)

 α err prob = 0.05

 Sample size group 1 = 3

 Sample size group 2 = 3

Output: Noncentrality parameter δ = 24.4948974

 Critical t = 2.7764451

 Df = 4

 Power (1-β err prob) = 1.0000000 (100%)

KKU505 Phage 8 VS VSCM13 helper phage

t tests - Means: Difference between two independent means (two groups)

Analysis: Post hoc: Compute achieved power 

Input: Tail(s) = Two

 Effect size d = 20 (25.2635)

 α err prob = 0.05

 Sample size group 1 = 3

 Sample size group 2 = 3

Output: Noncentrality parameter δ = 24.4948974

 Critical t = 2.7764451

 Df = 4

 Power (1-β err prob) = 1.0000000 (100%)

3. Have the authors made all data underlying the findings in their manuscript fully available?

 Reviewer #1: No

4. Is the manuscript presented in an intelligible fashion and written in standard English?

 Reviewer #1: Yes

5. Review Comments to the Author

Reviewer #1: The paper entitled “Production and characterization of antibody against Opisthorchis viverrine via phage display and molecular dynamics simulations” addresses on the one hand the issue of safe disposal of hybridoma instability and on the other hand shows the utility of phage display and MD simulation to produce and characterize the KKU505 antibody.

The study mentions in the title and in the discussion an MD simulation of 15 ns, but neither in the methods nor in the Supporting Information the parameters used for MD simulations are shown. Were the simulations performed in implicit or explicit solvent, which force field was used, which thermostat, barostat, ... Apart from that also no MD simulation data is shown (RMSD plot, clustering...). Are 15 ns enough to draw any conclusions, especially when the complex structure was determined by docking, a longer relaxation time of the complex might be necessary. 

Response: Thank you so much for your suggestions. First, I have to apologize that makes you some confusion. This manuscript entitled “Production and characterization of antibody against Opisthorchis viverrine via phage display and molecular simulations” is focused on molecular simulation to illustrate the 3D modelling and docking for better understanding and show how possible my phage display antibody interacts with the antigen. We also did the molecular dynamic simulation and we agree with your mention above that the molecular dynamic simulation for 15 ns timescale, not enough for showing the results, therefore, we discussed this molecular dynamic simulation in the discussion part for a preliminary result which needs more analysis.

Finally, we put some information about this molecular dynamic simulation and discuss the timescale problem as shown in the red letter on page 11. There are

line 353-354 “This was calculated using the YASARA software package [48] with the AMBER14 force field for the duration of 15 nanoseconds timescale (unpublished data).”

line 361-362 “However, the analysis of protein-protein interaction as investigated through molecular simulation may needs more analysis timescale.”

Additionally, the antibody structure was also modelled, which introduces another uncertainty. Recent literature described the timescales of CDR loop rearrangements in combination with also the fast VH-VL interdomain orientations, which also influence the shape of the paratope and might also have an effect on the resulting docked complex. Additionally, please also take the following review about allergen-antibody interactions into account.

• https://doi.org/10.1080/19420862.2020.1744328

• https://www.ncbi.nlm.nih.gov/pmc/articles/PMC7492603/

• https://www.nature.com/articles/s42003-020-01319-z.

Please discuss the timescale problem and also please mention if the template used for the antibody homology model was based on an unbound Fab structure. If yes please also discuss the problems of potential crystal packing effects, which can distort the CDR loops, which are involved in binding.

• https://www.tandfonline.com/doi/full/10.1080/19420862.2019.1618676

Please comment to this.

Response: We have already reviewed your given topics mentioned above, and also discussed more details about these topics in the discussion part as shown in the red letter, there are;

on page 11, line 345-350 

“As for antibody modeling, Fernández-Quintero and co-workers [47] suggested the importance of Fab structure and special care should be taken into consideration. Particularly, the antibody crystal structure without the antigen present was found to bind to the tail region of the Fab. In this study, the modeled Fab structure in the absence of the antigen was used whereby the best antibody-antigen complex model was selected by avoiding antigen binding to the tail of the Fab structure.”

on page 11, line 354-361 

“Findings from molecular dynamics simulation conducted by Fernández-Quintero et al. [49] shows that the CDR loop ensemble of antibodies exert its effects on the paratope states in solution. Furthermore, it was found that CDR loops can shift the relative VH and VL orientation, the elbow-angle structure distributions, and the dynamic simulation [50]. Thus, these aforementioned results show that the antibody CDR loops plays an important role in the antibody properties. These results suggested that the antibody CDR loops is an important region that should be considered in antibody structure design as well as in molecular simulation.”

on page 11, line 368-372 

“Moreover, Pomés et al. [52] also demonstrated the benefits of computational methods in conjunction with experiments for studying allergen-antibody interaction and allergens antibody binding epitopes whereby antibody phage display technology was used in combination with a computational algorithm for identifying the allergen surface that mimics conformational epitopes.”

Then, the new references from your suggestion were add in the reference list in this manuscript as shown in the red letter, there are;

on page 17-18, line 596-612

“47. Fernandez-Quintero ML, Kraml J, Georges G, Liedl KR. CDR-H3 loop ensemble in solution - conformational selection upon antibody binding. MAbs. 2019;11(6):1077-88. Epub 2019/06/01. doi: 10.1080/19420862.2019.1618676. PubMed PMID: 31148507; PubMed Central PMCID: PMCPMC6748594.

48. de Groot BL, van Aalten DM, Scheek RM, Amadei A, Vriend G, Berendsen HJ. Prediction of protein conformational freedom from distance constraints. Proteins. 1997;29(2):240-51. Epub 1997/11/05. PubMed PMID: 9329088.

49. Fernandez-Quintero ML, Heiss MC, Pomarici ND, Math BA, Liedl KR. Antibody CDR loops as ensembles in solution vs. canonical clusters from X-ray structures. MAbs. 2020;12(1):1744328. Epub 2020/04/09. doi: 10.1080/19420862.2020.1744328. PubMed PMID: 32264741; PubMed Central PMCID: PMCPMC7153821.

50. Fernandez-Quintero ML, Pomarici ND, Math BA, Kroell KB, Waibl F, Bujotzek A, et al. Antibodies exhibit multiple paratope states influencing VH-VL domain orientations. Commun Biol. 2020;3(1):589. Epub 2020/10/22. doi: 10.1038/s42003-020-01319-z. PubMed PMID: 33082531; PubMed Central PMCID: PMCPMC7576833.”

on page 18, line 618-621

“52. Pomes A, Mueller GA, Chruszcz M. Structural Aspects of the Allergen-Antibody Interaction. Front Immunol. 2020;11:2067. Epub 2020/09/29. doi: 10.3389/fimmu.2020.02067. PubMed PMID: 32983155; PubMed Central PMCID: PMCPMC7492603.”

6. PLOS authors have the option to publish the peer review history of their article (what does this mean?). If published, this will include your full peer review and any attached files.

Do you want your identity to be public for this peer review? For information about this choice, including consent withdrawal, please see our Privacy Policy.

Reviewer #1: No

Other

In addition, we edited some information for making a clear understanding as shown in the red letter. There are

on page 2, line 7 and 14 “dynamic” were removed 

on page 4, line 85 “dynamic” was removed 

on page 4, line 90 “simulation” was instead of “dynamic” 

on page 7, line 194 “can be divided” was instead of “divides”

on page 7, line 195 “major” was instead of “studying”

on page 11, line 339 “dynamic” was removed 

on page 11, line 351 “dynamic” was removed

on page 11, line 351 “simulation” was removed

on page 11, line 352-353 “time span 15 ns” was removed

Furthermore, we edited some reference cite for making more correct as shown in the red letter. There are

on page 3, line 50 “The study of Worasith et al.” was instead of “The study by Chanika Worasith”.

on page 3, line 63 “et al.” was add after the last name of the first author (Worasith).

on page 7, line 197 “Thana et al.” was instead of “Duanggamon”.

on page 10, line 291 “Thana et al.” was instead of “Duanggamon”.

on page 10, line 314 “the work of Worasith [15]” was instead of “Chanika experiment”.

on page 11, line 364 “Experiments conducted by Lousa et al. [51]” was instead of “The experiment by Diana Lousa”.

on page 12, line 374 “Banchob” was deleted.

on page 12, line 374 “Sasithorn” was deleted.

---

## [Decision Letter · Decision Letter 1]

8 Mar 2021

Production and characterizationof antibody against Opisthorchis viverrini via phage display and molecular simulation

PONE-D-20-29938R1

Dear Dr. Jumnainsong,

We’re pleased to inform you that your manuscript has been judged scientifically suitable for publication and will be formally accepted for publication once it meets all outstanding technical requirements.

Kind regards,

Theerapong Krajaejun, M.D.

Academic Editor

PLOS ONE

Additional Editor Comments (optional):

The authors have properly addressed the reviewer's concerns, and the manuscript is now suitable for publication.

Reviewers' comments:

Reviewer's Responses to Questions

**Comments to the Author**

1. If the authors have adequately addressed your comments raised in a previous round of review and you feel that this manuscript is now acceptable for publication, you may indicate that here to bypass the “Comments to the Author” section, enter your conflict of interest statement in the “Confidential to Editor” section, and submit your "Accept" recommendation.

Reviewer #1: All comments have been addressed

2. Is the manuscript technically sound, and do the data support the conclusions?

Reviewer #1: Yes

3. Has the statistical analysis been performed appropriately and rigorously? 

Reviewer #1: (No Response)

4. Have the authors made all data underlying the findings in their manuscript fully available?

Reviewer #1: Yes

5. Is the manuscript presented in an intelligible fashion and written in standard English?

Reviewer #1: Yes

6. Review Comments to the Author

Reviewer #1: (No Response)

7. PLOS authors have the option to publish the peer review history of their article (what does this mean?). If published, this will include your full peer review and any attached files.

Reviewer #1: No

---

## [Editor Report · Acceptance letter]

11 Mar 2021

PONE-D-20-29938R1 

Production and characterization of antibody against *Opisthorchis viverrini* via phage display and molecular simulation 

Dear Dr. Jumnainsong:

I'm pleased to inform you that your manuscript has been deemed suitable for publication in PLOS ONE. Congratulations! Your manuscript is now with our production department. 

Kind regards, 

on behalf of

Dr. Theerapong Krajaejun 

Academic Editor

PLOS ONE